

# Estimating Uncertainties in the SBUV Version 8.6 Merged Profile Ozone Dataset

Stacey M. Frith[1], Richard S. Stolarski[2], Natalya A. Kramarova[1], Richard D. McPeters[3]

[1]Science Systems and Applications, Inc., Lanham, MD, US
[2]Dept of Earth and Planetary Sciences, Johns Hopkins University, Baltimore, MD, USA
[3]NASA Goddard Space Flight Center, Greenbelt, MD, USA

*Correspondence to*: Stacey M. Frith (Stacey.frith@nasa.gov)

**Abstract.** The combined record of total and profile ozone measurements from the Solar Backscatter Ultraviolet (SBUV) and SBUV/2 series of instruments, known as the SBUV Merged Ozone Data (MOD) product, constitutes the longest satellite-based ozone time series from a single instrument type, and as such plays a key role in ozone trend analyses.
Following the approach documented in Frith et al. (2014) to analyze the merging uncertainties in the MOD total ozone record, we use Monte Carlo simulations to estimate the potential for uncertainties in the calibration and drift of individual instruments in the profile ozone merged data set. We focus our discussion on the trends and associated merging uncertainty since 2001 in an effort to verify the start of ozone recovery as predicted by chemistry climate models. We find that merging uncertainty dominates the overall estimated uncertainty when considering only the 15 years of data since 2001. We derive trends versus pressure level for the MOD data set that are positive in the upper stratosphere as expected for ozone recovery. These trends appear to be significant when only statistical uncertainties are included, but become not significant at the 2σ level when instrument uncertainties are accounted for. However, when we use the entire data set from 1979 through 2015 and fit to the EESC (equivalent effective stratospheric chlorine) we find statistically significant fits throughout the upper stratosphere at all latitudes. This implies that the ozone profile data remain consistent with our expectation that chlorine is the dominate ozone forcing term.

## 1 Introduction

The solar backscatter ultraviolet (SBUV) series of instruments provides a 40+ year data record of broadly resolved vertical ozone profiles on a global scale. We recently reported on our updated Merged Ozone Data (MOD) record of integrated total column SBUV measurements (Frith et al., 2014). Here we extend the record by considering the ozone profile measurements in layers from 25–1 hPa, where SBUV provides the best vertical resolution (Kramarova et al., 2013a, Bhartia et al., 2013). The SBUV record comprises data from 9 instruments (Nimbus 4 BUV, Nimbus 7 SBUV and SBUV/2s on NOAA 9, 11, 14, 16, 17, 18, 19), providing ozone measurements over an era of changing chlorine levels and changing stratospheric climate. In



order to isolate these signals from natural ozone variability, a single coherent data set is required. To this end, instruments in the series were cross-calibrated at the wavelength level using overlapping measurements collected within defined spatial and temporal limits (DeLand et al., 2012). Ozone was then derived for each instrument using the Version 8.6 retrieval algorithm to produce the measurement time series used to create the MOD data set (McPeters et al., 2013). While this approach
minimized differences among instruments, Frith et al. (2014) showed that small remaining offsets and drifts between measurements contributed to the uncertainty in the total ozone SBUV MOD record.

Profile ozone measurements are inherently noisier than column ozone measurements, and generally show larger variations between measurements (e.g. Kramarova et al., 2013b; Hassler et al., 2014; Tummon et al., 2015; Hubert et al., 2016). For the SBUV profile measurements, different wavelengths are sensitive to the ozone concentration at different pressure levels
and wavelength is being used to "scan" the profile. Wavelength-dependent calibration errors tend to cause ozone errors that oscillate in the vertical. Thus profile ozone measurements are much more sensitive to wavelength calibration and instrument issues while these errors tend to cancel in total ozone.

Although profile measurements have an inherently larger uncertainty than total ozone measurements (at least for SBUV), these data are critical in the search for indicators of the recovery of ozone. Model studies indicate that the expected recovery
of ozone from the impacts of chlorine and bromine compounds will be latitude and altitude dependent (e.g. Li, et al., 2009). To that end, a number of recent studies have indicated statistically significant ozone increases since the late 1990s based on merged ozone profile records, suggesting recovery from the earlier ozone decline attributed to ozone depleting substances (ODSs; e.g., WMO, 2014; Tummon et al., 2015; Harris et al., 2015; Steinbrecht et al., 2017). However, a full characterization of the uncertainties associated with the merging process of data from multiple instruments is not generally
available. It is important that not only the individual instrument uncertainties be taken into account, but uncertainties arising from the merging process itself must also be accounted for before the merged data can be properly interpreted. Such uncertainties result from individual instrument uncertainties (absolute calibration, drift, other systematic errors) but also from differences in measurement technique, spatial and temporal resolution, and native vertical coordinate systems of the merged records (e.g. Damedeo et al, 2014; Hassler et al., 2014; Sofieva et al., 2015). How to propagate such uncertainties and assess
their impact on derived trends and other long-term signals is an outstanding question within the community (e.g. WMO 2014, Harris et al., 2015).

In this work we estimate the merging uncertainty in the SBUV profile MOD record using a Monte Carlo approach similar to that presented in Frith et al. (2014) for total ozone. In the following sections we describe the SBUV measurements and analyze the consistency between the individual instrument data sets. We then describe the merged data product and
summarize our approach to estimating potential long-term drifts in the data set. Using a multiple linear regression model we compute profile ozone trends, focusing on the ozone changes since 2000, a period when models project ozone beginning to recover and the SBUV/2 data are most reliable. We evaluate two sources of uncertainty: the statistical uncertainty resulting





from real atmospheric variability and an imperfect regression model fit to the data; and the uncertainty in merging the data from multiple instruments because of possible offsets or drifts in calibration. Finally we compare our results with trends derived from an independent merged record based on the same SBUV instrument data, known as the NOAA Cohesive data set (Wild et al., 2017). The NOAA Cohesive data set represents a different but reasonable approach to merging the same raw

data, and as such, our estimates of merging uncertainty should encompass differences in the derived trends between the data sets.

## 2 Data

The SBUV instrument series and most recent Version 8.6 data processing have been described in detail in a series of publications (Deland et al., 2012; McPeters et al., 2013; Bhartia et al., 2013). The data have been assessed and compared to

independent measurements by Labow et al. (2013) and Kramarova et al. (2013b).  Here we focus on the most recent updates, including data from NOAA 19 that were not included in the aforementioned studies.   Figure 1 shows an update of the BUV instrument orbit drift history since 2000. Here we plot the local time at which each satellite crosses the equator as a function of time. SBUV instruments ideally operate in late morning-early afternoon sun synchronous orbits such that measurements are made at small solar zenith angles and at the same local time each orbit. NOAA satellites are launched into orbits that

slowly drift toward the terminator, and several satellites have drifted through the terminator, thus making both afternoon and morning measurements (see NOAA 16 in Figure 1). The afternoon and morning orbital segments are denoted by the suffix "pm" and "am" in the following discussion (as in NOAA 16_pm and NOAA 16_am). The first NOAA instruments (NOAAs 9/11/14) underwent more pronounced orbital drift and the associated data quality during portions of these records is notably reduced (DeLand et al., 2012; Kramarova et al., 2013b; McPeters et al., 2013).

We follow the same data selection criteria as used in Frith et al. (2014) based on prior data quality assessments. Namely, we only use measurements made when the satellite equator crossing time is between 8 am and 4 pm to avoid issues with drifting orbits, with the exception of NOAA 11_pm to avoid a data gap. We do not include NOAA 9 data due to quality issues. Otherwise we retain the Tier 1 and Tier 2 instrument quality designations, with the aforementioned instruments in drifting orbits generally assigned as Tier 2. Tier 2 instruments are of lesser quality than Tier 1 but are still considered useful within

the record, and include NOAA 11_am, NOAA 14_pm, NOAA 14_am, and NOAA 16_am. While we include Tier 2 data when creating the long-term MOD data record, we account for the varying data quality in our uncertainty estimates. Additionally for the profile data set, measurements are removed for at least a year after the volcanic eruptions of El Chichon and Mt. Pinatubo (Bhartia et al., 2013), and no NOAA 9 profile data are used (limited NOAA 9 data were used in the total ozone MOD to fill data gaps).

Data from the next generation Ozone Mapping and Profiler Suite (OMPS) Nadir Profiler (NP) instrument, launched in October 2011 on the Suomi-NPP satellite, and subsequent planned missions on the JPSS (Joint Polar Satellite System) series will continue the nadir profiler measurements for decades to come.  The Suomi-NPP is in a stable early afternoon orbit while





the NOAA-19 satellite has begun to drift towards the terminator. The current merged data set does not include OMPS NP measurements but we anticipate these data will be added soon, as the orbit for NOAA-19 approaches the 4pm ECT cutoff and the OMPS data are reprocessed with an improved long-term calibration (Version 2).

Kramarova et al. (2013b) presented a complete validation of the individual SBUV instrument measurements relative to

satellite and ground-based independent data sources. Figure 2 shows updated mean bias comparisons between Aura Microwave Limb Sounder (MLS; daytime measurements only) and SBUV NOAA 16/17/18/19 averaged in three broad latitude bands. Here we use V4.2 MLS data and include NOAA 19 SBUV/2 data as well as updated revisions of version 8.6 for NOAA 16/17/18 SBUV/2 data, but the conclusions remain the same as in Kramarova et al. (2013b). Relative to Aura MLS, SBUV biases are largely within 5%. The SBUV instruments in afternoon orbits (NOAAs 16_pm/18/19) show a

distinct pattern with SBUV lower than MLS in the upper stratosphere, higher than MLS in the middle stratosphere and then lower than MLS in the lower stratosphere. The two SBUV instruments in morning orbits (NOAA 16_am and NOAA 17) show a qualitatively similar but less pronounced pattern with respect to MLS. Using MLS as a transfer standard, the morning orbit SBUV measurements are smaller in the middle stratosphere and greater in the upper stratosphere than the afternoon orbit measurements. This pattern is generally consistent with diurnal variations in ozone observed from ground-based

microwave measurements at Mauna Loa (Parrish et al., 2014).

Figure 3 shows the drifts of the SBUV instruments relative to Aura MLS over the time period when both instruments are making measurements, computed after removing the respective seasonal cycles from each data record. The drifts are generally within 5% dec$^{-1}$. The largest drifts (> 5% dec$^{-1}$ in some cases) are those for NOAA 16 in both the morning and afternoon orbits. Both of these records overlap MLS by only 30-35 months. The longer overlaps for NOAAs 17/18/19 (80+

20   months) show significantly less drift with respect to Aura MLS. Although the 2σ uncertainty bounds in Figure 3 indicate that these larger drifts for NOAA 16 are significant, they do not include consideration of autocorrelation and are almost certainly an artifact of the short data overlap. Kramarova et al. (Figure S19, 2013b) use NOAA 17 data to test the effects of short overlaps when comparing to ground-based microwave data from Mauna Loa by varying the length of overlap from 18 to 114 months. The relative drift does not stabilize until the overlap is ~70 months or longer.

Thus far we have focused on the SBUV data since 2000. In Figure 4 we examine inter-instrument consistency over the entire MOD record, starting in 1970. Here we show examples of the full time series of the individual SBUV instruments at three pressure levels for the latitude band from 35 to 50N. The data are plotted as the anomalies from the N7 SBUV 1979-1981 seasonal cycle.

The records from the various satellite SBUV instruments present a consistent picture of ozone variation over time. In the 2.5

to 1.6 hPa layer we see a rapid decline of about 15% between 1980 and about 2000 with a leveling off after 2000 and perhaps a slight increase. In the 6.4 to 4 hPa layer we see a smaller decrease. In the lower 16 to 10 hPa layer we see a much smaller decrease with a strong deviation in the seasonal cycle compared to the reference period of 1979 to 1981. This deviation in seasonal cycle was present in the higher layers, but not as pronounced. These seasonal differences may in part be a real response to long-term chlorine changes (e.g., Stolarski et al., 2012 and references therein).



Despite the overall coherence among the individual instrument records, offsets and drifts between the instruments are apparent. The largest differences occur in the mid to late 1990s and again in the mid-2000s associated with the Tier 2 NOAA 11_am, N14_pm and N14_am data. Smaller differences can be seen among the Tier 1 instruments as well.

## 3 Analysis

In this section we focus on the quantitative analysis of the SBUV time series with emphasis on how the uncertainties in the measurements feed into the overall uncertainty in the merged ozone record. We use the same approach described by Frith et al. (2014), in which uncertainties have been constructed for the merged total ozone data record. Profile ozone differences among the individual SBUV instrument measurements are larger than was the case for total ozone, but the uncertainty issues are quite similar. That is, there are two major sources of uncertainty in combining the measurements from multiple satellite

instruments. The first source of uncertainty is related to absolute calibration offsets between instruments, while the second stems from possible calibration drift over the lifetime of the instruments. We attempt to quantify these uncertainties, and then model their time dependence using a Monte Carlo approach to estimate their impact on the long-term variability in the ozone profile measurements.

### 3.1 Monte Carlo Uncertainty Model Parameters

Figures 2, 3, and 4 show ozone profile measurements from individual SBUV instruments that are generally similar during periods of overlap but include a range of inter-instrument offsets and drifts. The individual instrument data sets were produced by the SBUV processing team after considering all of the known issues with respect to the calibration of each instrument (DeLand et al. 2012). Despite the best efforts of this team to obtain the most accurate possible calibration of each instrument, differences remain that depend on latitude and altitude. To construct the MOD data set we average the

individual data records during periods of overlap of two or more instruments within the 8 am to 4 pm ECT boundary.

One could argue that we could simply adjust offset differences at each latitude and altitude so that we had one continuous data set with little or no relative offset uncertainty. However, the existing data from each instrument had what the processing team deemed the best possible calibration. The remaining differences in ozone measurement during overlap periods are due to factors that are not understood at this time. This means that it is not clear how to adjust data to remove these offsets. An

offset between two instruments during their period of overlap could result from the calibrations being different, but it could also result from a drift over time of one or both of the instruments from its initial calibration. The issues of offset and drift are thus inextricably linked. Therefore, instead of arbitrarily adjusting differences between two instruments during their period of overlap, we have chosen to consider the offsets and drifts as part of the uncertainty in "stitching" together the results from the individual satellite instruments to form our merged data set. In the absence of a standardized reference

calibration source, we are in essence using the collection of SBUV inter-instrument offsets and drifts to define an SBUV-system uncertainty, in an effort to account for both relative and absolute uncertainties.



To estimate offset and drift uncertainty for the SBUV profile data records we use the same approach as described in Frith, et al. (2014). We compute the mean bias and drift for the overlap pairs in each 5-degree latitude bin, and use the collection of these values to determine the distribution of offsets and drifts for Tier 1 and Tier 2 instruments (see Frith et al., 2014, Figures 5 and 6 for a detailed explanation of Tier 1 and Tier 2 instruments). We scale the offsets and drifts by the square root of 2 to

distribute the relative offset/drift between instrument pairs.

Figure 5 shows the distribution of the absolute value of the biases for each instrument pair/latitude zone and the root mean square of the distribution to be used in the Monte Carlo simulations. As before, we compute the inverse error weighted root mean square of the individual biases. The inverse weighting allows us to account for length of overlap and autocorrelation in the differences for each instrument pair. That is, biases/drifts computed from longer overpass periods have greater weighting,

while comparatively high autocorrelation reduces the weight. However, we do not account for correlation in latitude, treating each band as an independent measure of the bias. In addition, with the larger differences in the profile data, we found the resulting distributions were more skewed than in total ozone. Relevant weighting and scaling is applied to the biases in the plotted distribution. Figure 6 shows the distribution of drifts between overlapping measurements. We require 24 months of overlap to compute drift, leaving only two Tier 2 instrument pairs with sufficient overlap, NOAA 11_am/NOAA 14_pm and

NOAA 16_am/NOAA 19 (in our total ozone analysis only 1 Tier 2 instrument pair was used). In this case, the distribution of drifts is sufficiently different for the two pairs that we treat them individually. That is, the N16_am drift is assigned based on the NOAA 16_am/NOAA 19 overlap (solid blue line in Figure 6), while the NOAAs 11_am/14_pm/14_am drifts are based on the NOAA 11_am/NOAA 14_pm relative drift (dashed blue line in Figure 6). Again relevant weighting and scaling is applied to the biases in the plotted distribution.

The root mean square of the bias is less than 2% for the Tier 1 instruments, and less than 3% for Tier 2. In general, high quality satellite based profile ozone observations agree to within ~5% (Figure 2; Kramarova et al., 2013b; Tummon et al., 2015; Hubert et al., 2016). Similarly, Kramarova et al. (2013b) found the drift of the higher quality SBUV records (our Tier 1) to be within 0.5 percent per year when compared to independent measurements, and we see similar size drifts between the individual Tier 1 SBUV measurements, with slightly larger drifts indicated for the NOAA-16am data. Hubert et al., (2016)

analyzed data from 14 limb and occultation sounders relative to ground-based reference data sets and also found most instruments were stable to within 0.5 percent per year against the ground reference in the middle and upper stratosphere, though the satellite to satellite drifts might be larger. Overall, with the exception of the large drift between NOAA-11_am and NOAA-14_pm, offset and drift between individual SBUV instruments are comparable to that found in other satellite-based profile ozone measurements.

**3.2 Monte Carlo Model Structure**

The offset and drift parameters derived in the previous section form the basis for our Monte Carlo simulation of how these uncertainties lead to uncertainties in trend determination. We start under the assumption that the data from each satellite has a calibration that is unbiased with respect to the other satellites and, to the best of our knowledge, the calibration does not




drift in time. We are thus assuming that any uncertainty could go in either direction. For each instrument used in the MOD we then randomly prescribe offset and drift uncertainty from assumed Gaussian distributions with 1-σ widths equal to the root mean square values shown in Figures 5 and 6. Simulated Tier 1 instrument uncertainties are drawn from the Tier 1 distributions and Tier 2 uncertainties from the Tier 2 distributions, thereby explicitly representing the varying uncertainty in

the individual records. We treat drift and offset in a two-step process. In the first step, we apply only the drift uncertainty. We then inter-calibrate the drifting records using two reference instruments, NOAA 11_pm and NOAA 17, following the process used in the algorithm (DeLand et al., 2012; Frith et al., 2014). Through this adjustment we induce the time dependence we expect from the internal calibration process and we remove instrument to instrument offset that is solely due to drift in one or more instruments, as this offset is also a component of the offset distribution and we want to avoid double

counting uncertainties. We then add the offset uncertainty to each instrument and average to get a single simulated time series. This procedure is repeated 10,000 times to form the distribution of potential error due to the merging process.

Figure 7 shows two examples of Monte Carlo simulations based on the same bias and drift parameters described above at 10-16hPa. The first example represents a sequential merging process, with each new instrument adjusted to match the previous instrument. In this case the errors accumulate as each new record is added. The second figure shows the shape of the

simulations from the Monte Carlo model used to represent the SBUV MOD record outlined above. Here the Monte Carlo model is structured to reflect the timing of the V8.6 internal calibration, which is based on two reference calibration periods, NOAA 11_pm in the early 1990s and NOAA 17 in the mid-2000s (DeLand et al., 2012). Nimbus 7 and NOAA 9 (not used in MOD) are calibrated to NOAA 11_pm, while all later instruments are calibrated using NOAA 17 as the reference. This means that the uncertainties in the SBUV records do not accumulate sequentially from one instrument to the next, but grow

forward and backward in time away from the two reference data sets. Note that if the data set were constructed by referencing the records to an early single base calibration, such as Nimbus 7 SBUV, the modelled uncertainties would then follow the first example in the left panel of Figure 7. In this case the data at the end of the record would have a relatively large uncertainty with respect to the data at the beginning of the record because the calibration would have been transferred through the Tier 2 instruments (NOAAs 11/14) in the middle of the record.

**3.3 Multiple Linear Regression Model**

Having established a merging uncertainty distribution as a function of time, we now must convert this uncertainty to an uncertainty in derived trends from the merged data set. To analyze long term variability we use a standard multiple linear regression model including terms for the seasonal cycle, Quasi-biennial Oscillation, 11-yr solar cycle, volcanic aerosols from the eruptions of El Chichon and Mt. Pinatubo, El Nino/Southern Oscillation and either a fit to Equivalent Effective

Stratospheric Chlorine (EESC) using the full record or linear fits to segments of the data after long-term solar and volcanic variations have been removed. The regression model and proxy data sources are described in detail in Frith et al. (2014), and data sources are given in the Data Availability section of this paper.





We first fit the original MOD time series to the regression model. The statistical uncertainty, defined as the uncertainty associated with the imperfect ability of the proxies to capture all variability in the data, is computed using a bootstrap analysis of the residual time series. We run 400 iterations, and correlation in the residual is accounted for through 1-year segment resampling (*Efron*, 1979). We then compute the merging uncertainty by similarly running each of the Monte Carlo

uncertainty simulations (shown in Figure 7b) though the regression model and calculating the standard deviation of the resulting regression coefficients. The total uncertainty is the combination of the statistical and merging uncertainty, computed as the root mean square of the individual uncertainties.

For this analysis, as in Frith et al. (2014), we use a simplified regression model fitting only the EESC and solar terms to the uncertainty simulations over the full time period, or a simple linear trend when fitting to 1979-1994 or 2001-2015 time

segments. However we compared these results with fits to the full model and found very little difference in the final uncertainty estimates. Table 1 gives the derived merging uncertainty as a function of layer for the EESC fit, converted to drift units based on the slope of the EESC curve from 2001 to 2015, and for the linear segment fit over the same period. The larger uncertainty associated with the linear trend parameter reflects the larger potential for merging uncertainties to alias into trends over the relatively short 2001-2015 period compared to lower potential aliasing onto the EESC functional form

over the full time period. The linear trend parameter uncertainty is less than the 6% dec$^{-1}$ ($2\sigma$) error used in the most conservative approach presented by Harris et al., (2015), but greater than the uncertainty derived by Steinbrect et al. (2017) based on the spread of individual trends reported from a set of six merged ozone records.

Figure 8 shows the derived trend since 2001 estimated from EESC and from a linear fit as a function of latitude for the upper stratospheric 1.6-1.0 hPa layer. The statistical uncertainty and total uncertainty are shown separately. The trend derived by

20 either method is positive and nearly independent of latitude at a value of about 2% dec$^{-1}$. Both are statistically significant at the $2\sigma$ level if uncertainty in the merging process is not included. After adding the merging process uncertainty, the trend obtained using the EESC fit to the entire data set still yields a statistically significant trend after 2001 at the $2\sigma$ level. However, the trend obtained by fitting a linear function to the data after 2001 is now not statistically significant at the $2\sigma$ level (although it is close).

Figure 9 shows results at 10.1–6.4 hPa. The trend post-2001 for the EESC fit is small at all latitudes and not statistically significant except at the most southerly latitudes shown. When the merging uncertainty is added, the results are not significant at any latitude. For a linear fit to the data since 2001, somewhat larger trends are obtained that are significant at higher latitudes in both hemispheres if the merging uncertainty is excluded. However, when merging uncertainty is included these trends are no longer statistically significant. Smaller trends at this pressure level are consistent with many previous

analyses going back to Randel et al. (1999).



### 3.4 Comparison with NOAA Cohesive Data Set

The NOAA Cohesive data set in an independently constructed merged ozone record based on the SBUV series of instruments (Wild et al., 2017). For our purposes, we can consider the NOAA Cohesive data set as another realistic rendition of the MOD with variations that are defined by the intra-instrument differences (via adjustments applied in NOAA

Cohesive). Figure 10 shows the time series of the differences between the MOD data set and the NOAA cohesive data set for two pressure layers as a function of time. Differences, shown as blue dots, are plotted for all latitude bands to show the range of variations for each month. Also shown are the differences between MOD and the individual SBUV instrument monthly zonal means (black dots), which by definition are non-zero when more than one SBUV instrument contributes to MOD. The $2\sigma$ uncertainty limits defined from the variability of the Monte Carlo simulations are denoted by the red lines

(Figure 7). To the extent that the MOD and NOAA cohesive data sets are two realizations of reasonably-constructed time series from the SBUV data, we would expect these differences to fall within our estimated uncertainty bounds for most or all of the time series. The only exception is in the mid-1990s, but this is simply a matter of timing. NOAA Cohesive uses NOAA 9 data in 1994, while we continue with NOAA-11_pm until the NOAA 14 data start in early 1995. The uncertainty estimate reflects the timing of data used in the MOD record, but the magnitude of increased error with the introduction of a

Tier 2 instrument is sufficient to cover the range of differences with NOAA 9 (also Tier 2). In the early portion of the record both merged data sets are based on Nimbus 7 SBUV. The first variation comes in 1989, when MOD uses the average of Nimbus 7 and NOAA 11_pm, while the NOAA Cohesive data set switches to NOAA 11_pm. Starting in mid-1990s, differences increase as more instruments come online, and potential for the data sets to diverge increases. Throughout the period both individual SBUV and NOAA Cohesive measurements are generally contained within the MOD $2\sigma$ variability.

Figure 11 shows the trends calculated from the MOD and the NOAA cohesive data sets for the latitude bands of $35^{o}$–$50^{o}$S and $35^{o}$–$50^{o}$N as a function of pressure. The trends were derived from both merged records using a linear fit to data after 2001, computed after long-term variations in solar cycle are removed based on an initial full fit to the data. The shaded areas show the statistical uncertainty from both fits, while the errors bars show the combined merged and statistical uncertainty for the trend. Note that the merging uncertainty was calculated specifically for the MOD merging process, but for comparison

purposes we assume the same merging uncertainty for NOAA cohesive in the figure. The vertical structure of the derived trends is notably different between MOD and NOAA cohesive, though in most cases the statistical errors do overlap, if just barely. However, when the MOD merging uncertainty is added, the combined errors encompass both results. The reasons for these trend differences are apparent in Figure 10, where the MOD data drifted downward compared to NOAA cohesive in the 4–2.5 hPa, leading to a smaller upward trend. The opposite is true for the 16–10 hPa layer where MOD drifted upward

compared to NOAA cohesive resulting in more positive trends. Although we are fitting to 15 years of post-2000 data, end effects can still lead to differences in the trend.

On the other hand, we can examine the fits to EESC using the entire data set from 1979 through 2015. Figure 12 shows the trends derived for the time period 2001 to 2015 from both the MOD and NOAA cohesive data sets using EESC to fit the



entire data sets. We can see here that the trends derived from the two merged data sets are nearly identical when the entire data set is used to determine the fits. We also see that the trends in the top two layers are statistically significant including the merging uncertainties.

## 4 Summary and Conclusions

We have presented an analysis of the uncertainties in constructing a merged ozone data set for profile measurements of ozone from the SBUV instruments. The analysis is similar to our previous work on total ozone measurements from SBUV (Stolarski and Frith, 2006; Frith et al., 2014). However, the profile measurements are inherently nosier and the inter-calibration of instruments on different satellites is more uncertain. We find that inclusion of uncertainty in constructing the instrumental record has a great impact on determining the statistical significance of trend results derived from time-series

regression models. For instance, fitting a linear trend to the data since 2001 results in trends that are not statistically significant at the $2\sigma$ level at nearly all latitudes and altitudes when uncertainties related to the merging procedure are included.

Despite the insignificance of the derived trends from a linear fit to the data from 2001, we derive significant results for trends in the upper stratosphere at all latitudes by fitting to EESC over the entire data record. This significance comes mainly from

the fitting to the downward trend prior to 2000. EESC is a modelled representation of the amount of chlorine and bromine in the atmosphere available to destroy stratospheric ozone at a given time and location, based on measurements of ozone depleting substances in the troposphere, age of air, and the fractional release rates of chlorine and bromine from various chemical constituents (Newman et al., 2007). Our best current understanding, as represented in chemistry climate models, predicts that stratospheric ozone varies linearly with EESC (e.g., Newman et al., 2009). Therefore when we fit a data record

using the EESC as a regression parameter, we are testing the degree to which the data are following the model predictions. A fit to a linear trend on the other hand is a test of whether the data are increasing or decreasing as a result of any forcing not explicitly included in the regression model. Ideally a fit to linear trend can be used to verify that ozone is recovering as expected from chemistry climate models (i.e. following the EESC functional form), or alternatively, to indicate that other factors, such as stratospheric cooling or other unexplained long-term variations may also be affecting the data. However, the

merging uncertainty on trends over the relatively short 15 year time period do not yet allow us to independently verify the ozone recovery rate predicted by the model. Nevertheless, the continued goodness of fit of the data to the EESC curve extended through 2015 provides strong evidence that chlorine is the major driver of the long-term changes in the ozone concentration.

*Data Availability*.   The most recent version of the Merged Ozone Dataset (MOD) is available at https://acd-ext.gsfc.nasa.gov/Data_services/merged/index.html. Included on this website are links to the individual SBUV instrument monthly zonal mean records, which are the input records used to construct MOD. The Merged Cohesive data set is available



for download at ftp://ftp.cpc.ncep.noaa.gov/SBUV_CDR/. Long term variability in the ozone profile record is represented by various proxy time series: (1) EESC data are from http://acdb-ext.gsfc.nasa.gov/Data_services/automailer/ (Newman et al., 2007); (2) solar cycle data are from the Penticton/Ottawa 10.7cm solar radio flux measurement record available at http://www.ngdc.noaa.gov/stp/solar/flux.html; (3) QBO data are available at http://www.geo.fu-berlin.de/en/met/ag/strat/produkte/qbo/index.html#access (Naujokat, 1986); (4) Multivariate ENSO Index (MEI) data are found at http://www.esrl.noaa.gov/psd/enso/mei/ (Wolter and Timlin, 1993; 1998); (5) Volcanic aerosols are from 2-d model simulations as described in Frith et al. (2014) and Stolarksi et al. (2006).

*Acknowledgements.* The authors would like to acknowledge the SBUV instrument team members for their work producing the Version 8.6 SBUV data. We also thank J. Wild and C. Long at the NOAA/NWS/NCEP Climate Prediction Center for their work constructing the Merged Cohesive data set. S. M. Frith is supported under NASA Contract NNG17HP01C.

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




**Table 1. 2σ uncertainty associated with EESC and linear trend proxy terms based on the standard deviation of fits to 10,000 Monte Carlo simulations. The total uncertainty is the root mean square of these values and the statistical uncertainty derived from the goodness of the regression model fit.**

| SBUV Pressure Layer (hPa) | $EESC_{2001-2015}$ % dec$^{-1}$) | Linear $Trend_{2001-2015}$ % dec$^{-1}$) |
|---|---|---|
| 1.6-1 | 0.8 | 2.3 |
| 2.5-1.6 | 1.4 | 3.6 |
| 4-2.5 | 1.8 | 4.7 |
| 6-4 | 1.3 | 3.8 |
| 10-6 | 1.4 | 4.0 |
| 16-10 | 1.3 | 3.4 |
| 25-16 | 0.7 | 1.9 |





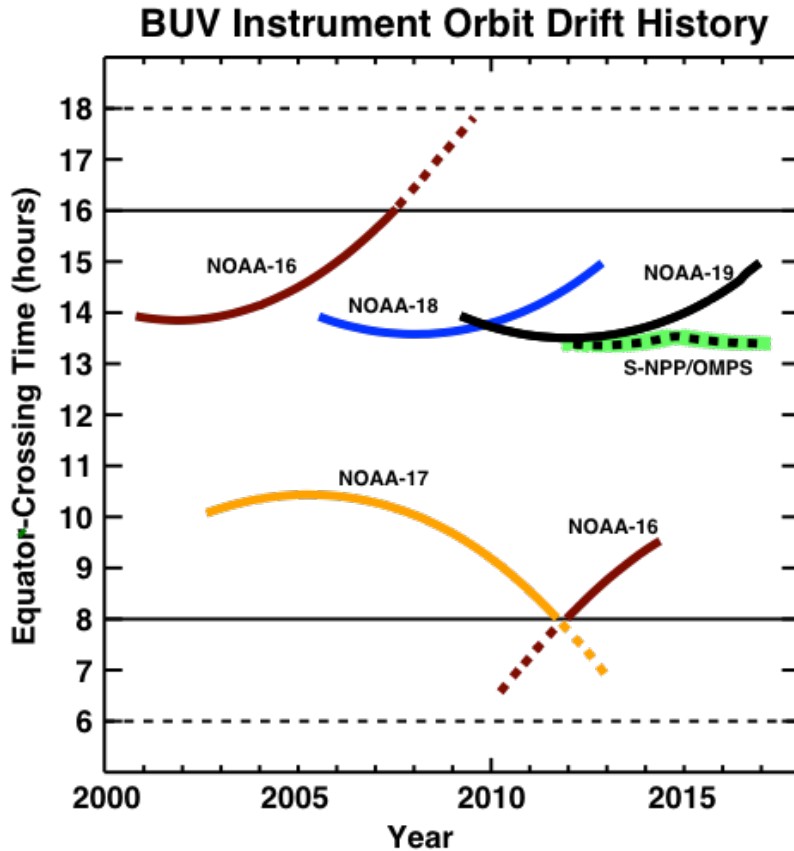

**Figure 1. Times series of Equatorial Crossing Time (ECT) for the series of SBUV instruments updated from 2000 through 2016. The terminator (6am and 6pm) ECTs are denoted by black dashed lines, and the 8am and 4pm ECTs are denoted by black solid lines. Thick color solid lines show segments of data included in MOD; dashed color lines show full extent of instrument record. S-NPP/OMPS is denoted by a solid line with dash overlay to indicate data that will be included in future MOD versions.**





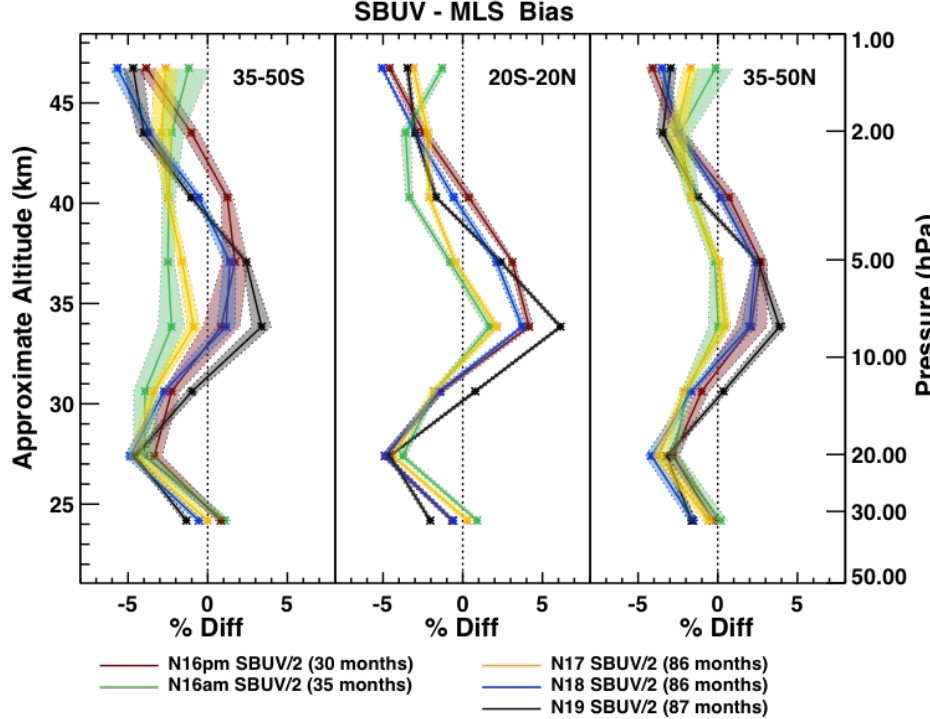

**Figure 2.** Mean bias of SBUV instruments (N16–N19) relative to Aura MLS in percent computed over respective overlap periods for each instrument. Results shown averaged in three broad latitude bands: 35–50° S; 20° S–20° N; and 35°–50° N. Shaded areas indicate 2σ deviations computed from the monthly means.





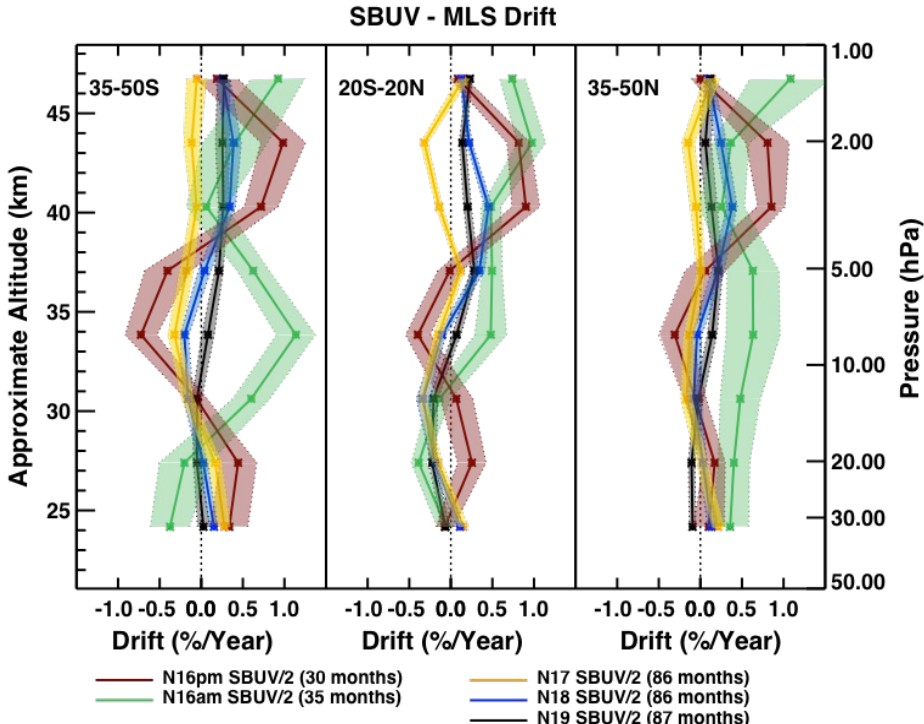

**Figure 3.** Drift of the NOAA SBUV instruments relative to Aura MLS over the time period of their overlap in % yr$^{-1}$ as a function of pressure level. Shaded areas indicate 2σ uncertainties not including autocorrelation.





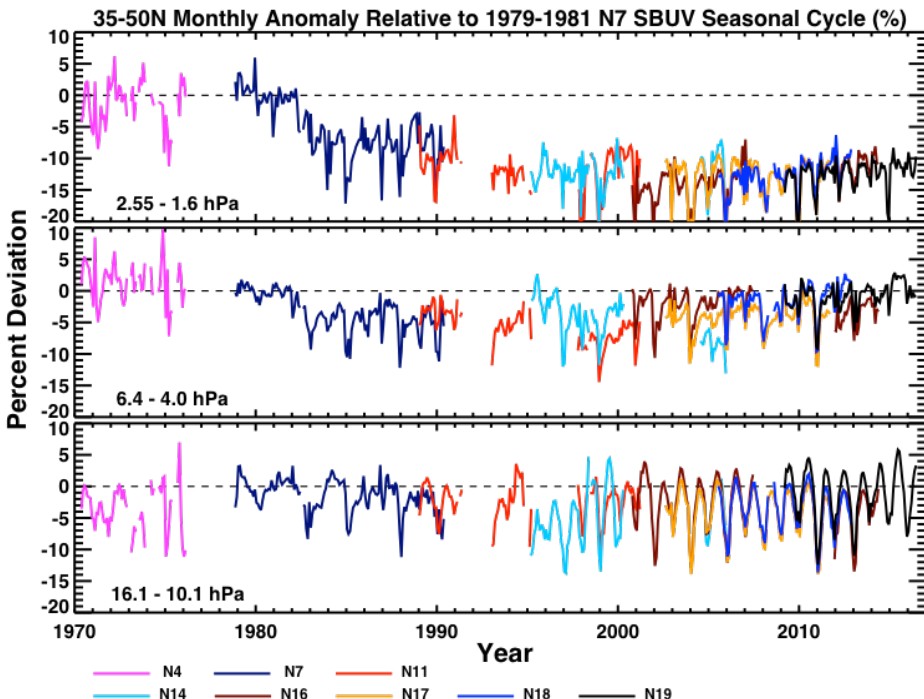

**Figure 4. Time series of ozone anomalies from individual SBUV records for three pressure levels. Anomalies are calculated from the 3-year 1979–1981 Nimbus 7 SBUV seasonal cycle. Data are averaged over the 35–50° N latitude band for: top panel 2.5–1.6 hPa, middle panel 6.4–4 hPa and bottom panel 16.1–10.1 hPa. The colors indicate the individual instrument records as indicated at the bottom of the figure.**



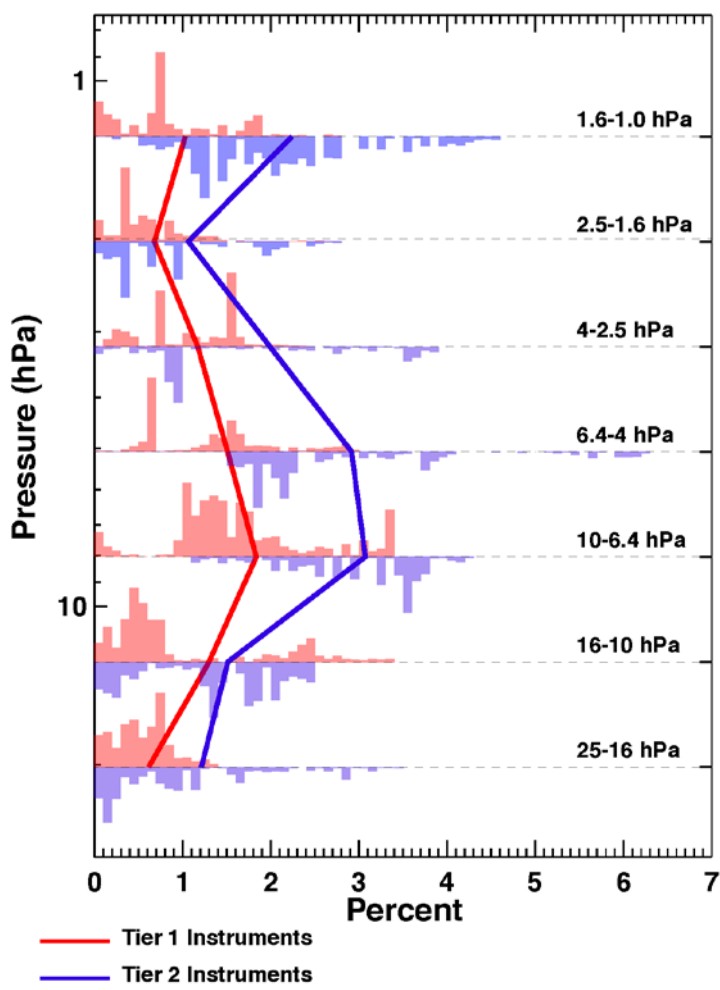

**Figure 5. Mean offset parameters as a function of pressure level derived from Tier 1 (Nimbus 7, NOAAs 11_pm/16_pm/17/18/19) and Tier 2 (NOAAs 11_am/14_pm/14_am/16_am) instrument overlaps used in Monte Carlo simulations to evaluate uncertainty of potential offsets and drifts on final merged ozone record. The parameters are the weighted root mean square of the collection of mean offsets computed in each 5° latitude bin and each overlapping instrument pair. The probability distribution for the Tier 1 instruments is shown by the red shaded histogram at each pressure level, while the probability distribution for the Tier 2 instruments is shown by the blue shaded histogram (shown upside down to separate it from the Tier 1 distribution).**



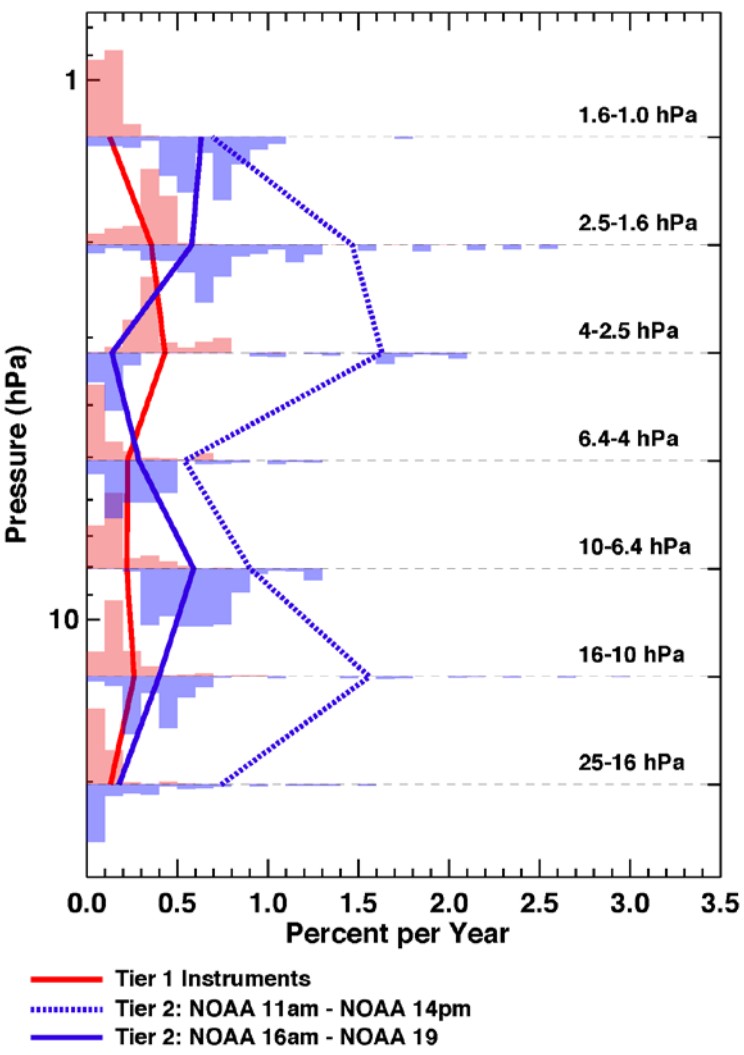

**Figure 6. Drift parameters as a function of pressure level derived from Tier 1 and Tier 2 instrument overlaps used in Monte Carlo simulations to evaluate uncertainty of potential offsets and drifts on final merged ozone record. For Tier 2 instruments, the NOAA 11_am to NOAA 14_pm relative drift is shown separately from that for NOAA 16_am to NOAA 19.**





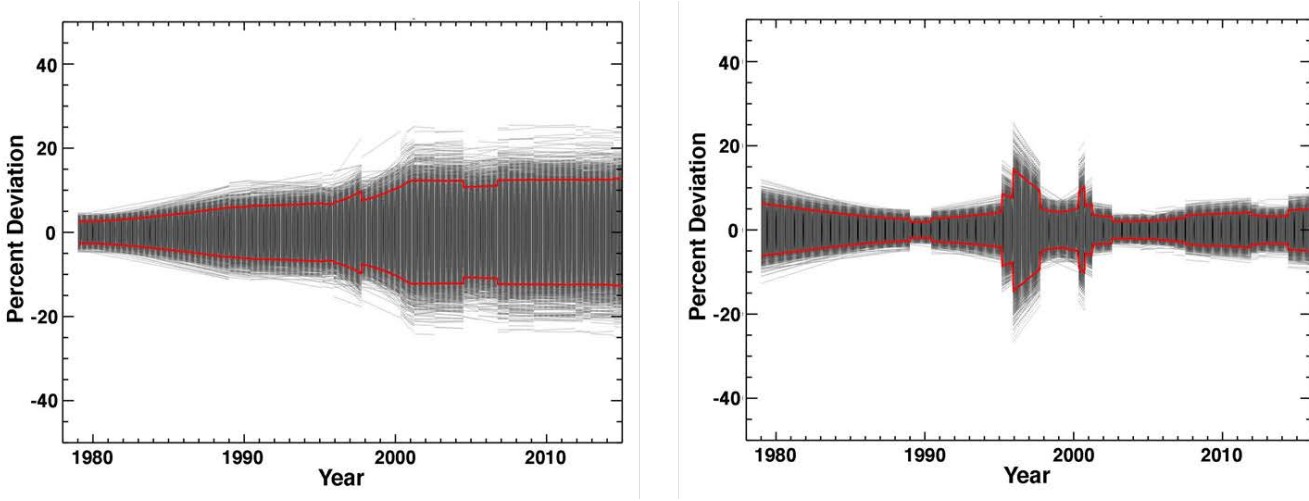

**Figure 7. Two examples of Monte Carlo simulations used to model propagation of uncertainty when merging data sets. The left panel shows a typical accumulation of uncertainty with each instrument added sequentially to the merged record. The second example simulations use the same distribution of offset/drift values, but the timing reflects the cross-calibration done within the V8.6 algorithm relative to two baselines, NOAA 11 in the early-1990s and NOAA 17 in the mid-2000s. In both cases, periodic reductions in the error spread occur during periods when two or more instruments are averaged.**





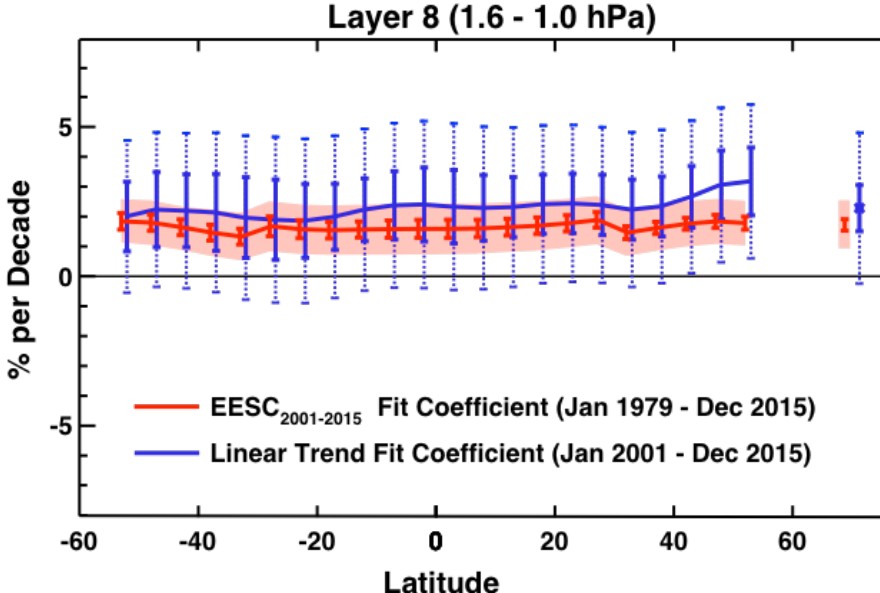

**Figure 8: Trend versus latitude for layer 8 (1.6–1.0 hPa) from 2001 to 2015 obtained by two methods. Red line and shading was obtained by fitting to EESC over entire time period of 1979 to 2015 and converting to slope of the EESC curve from 2001 to 2015. Solid vertical lines indicate 2σ uncertainty due to data variability. Red shading indicates uncertainty including impact of merging uncertainty. Blue line shows the trend obtained by fitting the data from 2001 to 2015 by a linear trend. Solid vertical bars are the 2σ uncertainty due to data variability while larger dashed vertical bars indicate addition of merging uncertainty. The points on the far right side are the values obtained for the 50°S–50°N average.**





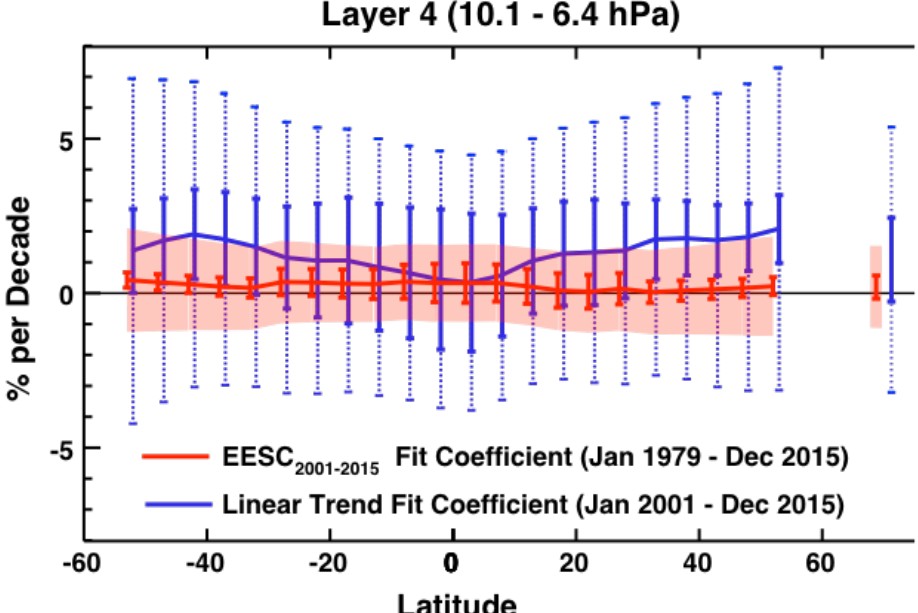

**Figure 9: Same as Figure 8 for 10.1–6.4 hPa.**





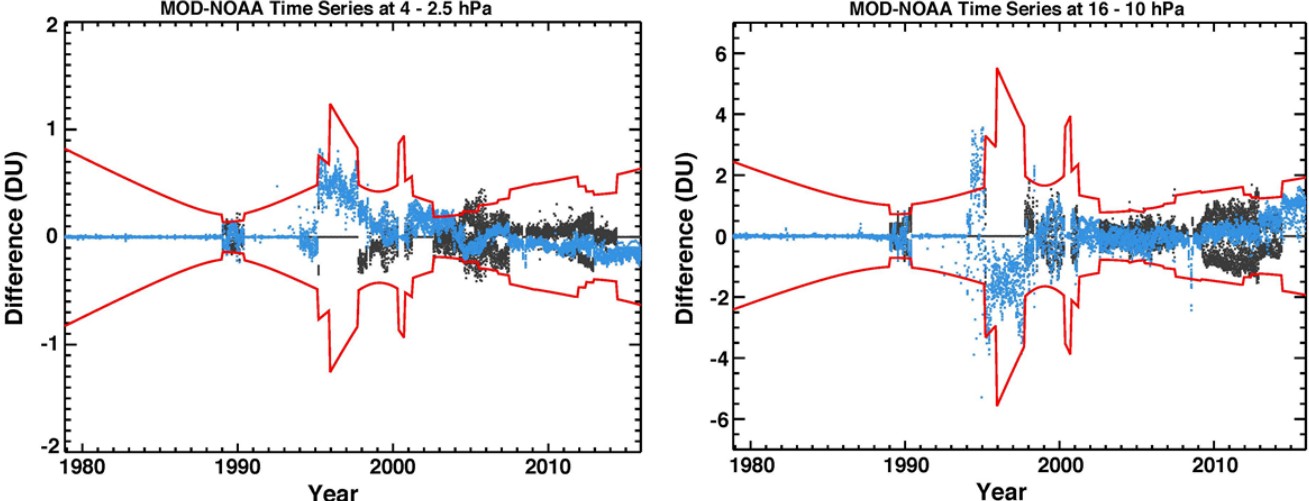

**Figure 10.** Time series showing MOD/NOAA cohesive data set differences at two pressure levels (4–2.5 hPa in left panel and 16–10 hPa in right panel). Black dots are MOD minus individual SBUV monthly mean differences, defined during times when more than one SBUV instrument contributes to MOD; Blue dots are MOD minus NOAA cohesive monthly mean differences. Both are generally contained within the 2σ variability. Differences in all latitude bands from 50°S to 50°N included. The red lines indicate 2σ variability of 10,000 Monte Carlo simulations.



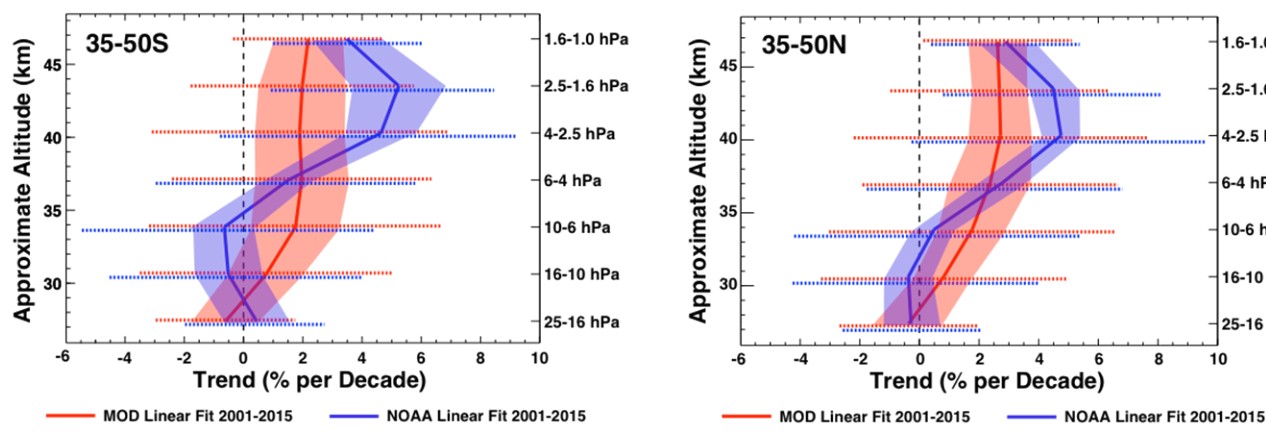

5    **Figure 11.** Linear trends fit from 2001 to 2015 using MOD (red) and NOAA Cohesive (blue) merged ozone records. Trends are in percent per decade and plotted as a function of pressure layers at 35–50°S (left panel) and 35°–50°N (right panel).





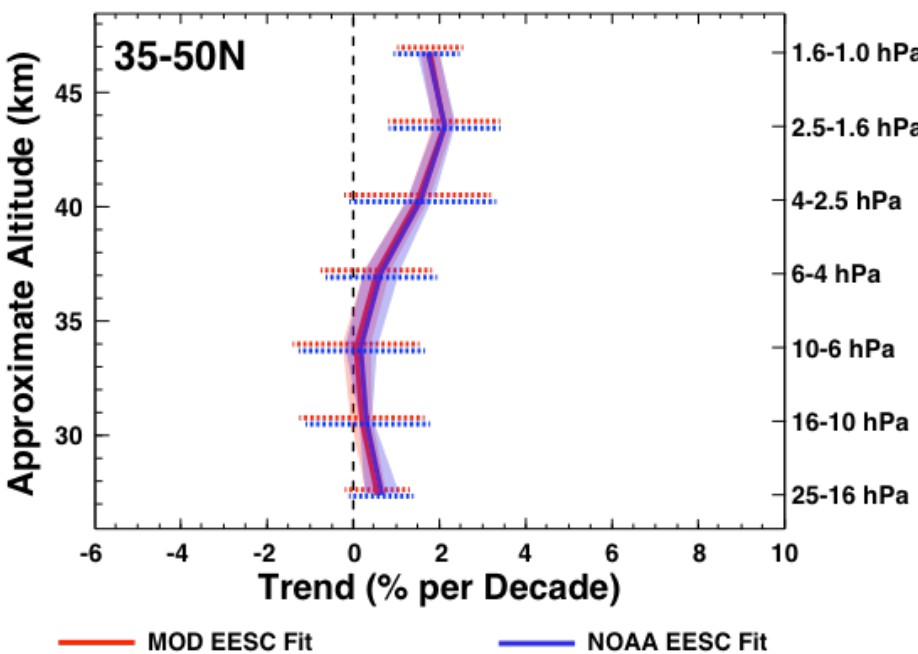

**Figure 12. Linear trends for the period 2001 to 2015 for latitude bin from 35°–50° N obtained by fitting EESC over entire time period from 1979 to 2015. Shading and uncertainties same as in Figure 11.**